# Macrophage Activation Markers, CD163 and CD206, in Acute-on-Chronic Liver Failure

**DOI:** 10.3390/cells9051175

**Published:** 2020-05-09

**Authors:** Marlene Christina Nielsen, Rasmus Hvidbjerg Gantzel, Joan Clària, Jonel Trebicka, Holger Jon Møller, Henning Grønbæk

**Affiliations:** 1Department of Clinical Biochemistry, Aarhus University Hospital, 8200 Aarhus N, Denmark; marlene@clin.au.dk (M.C.N.); holgmoel@rm.dk (H.J.M.); 2Department of Hepatology & Gastroenterology, Aarhus University Hospital, 8200 Aarhus N, Denmark; RAGANT@rm.dk; 3European Foundation for the Study of Chronic Liver Failure (EF-CLIF), 08021 Barcelona, Spain; JCLARIA@clinic.cat (J.C.); jonel.trebicka@efclif.com (J.T.); 4Department of Biochemistry and Molecular Genetics, Hospital Clínic-IDIBAPS, 08036 Barcelona, Spain; 5Translational Hepatology, Department of Internal Medicine I, Goethe University Frankfurt, 60323 Frankfurt, Germany

**Keywords:** macrophage, scavenger receptor, sCD163, sCD206, inflammation, chronic liver disease, acute decompensation of cirrhosis, liver cirrhosis, acute-on-chronic liver failure, ACLF

## Abstract

Macrophages facilitate essential homeostatic functions e.g., endocytosis, phagocytosis, and signaling during inflammation, and express a variety of scavenger receptors including CD163 and CD206, which are upregulated in response to inflammation. In healthy individuals, soluble forms of CD163 and CD206 are constitutively shed from macrophages, however, during inflammation pathogen- and damage-associated stimuli induce this shedding. Activation of resident liver macrophages viz. Kupffer cells is part of the inflammatory cascade occurring in acute and chronic liver diseases. We here review the existing literature on sCD163 and sCD206 function and shedding, and potential as biomarkers in acute and chronic liver diseases with a particular focus on Acute-on-Chronic Liver Failure (ACLF). In multiple studies sCD163 and sCD206 are elevated in relation to liver disease severity and established as reliable predictors of morbidity and mortality. However, differences in expression- and shedding-stimuli for CD163 and CD206 may explain dissimilarities in prognostic utility in patients with acute decompensation of cirrhosis and ACLF.

## 1. Introduction: Macrophages, Inflammatory Liver Diseases, and ACLF 

Macrophages play a significant role in acute and chronic inflammatory liver diseases being involved in both liver disease development and progression but also disease resolution. Liver cirrhosis is the cardinal endpoint of chronic inflammatory liver diseases where especially development of portal hypertension increases the risk of complications. Acute decompensation (AD) represents the classical complications in patients with liver cirrhosis e.g., variceal bleeding, ascites with hepatorenal syndrome, and hepatic encephalopathy, which has a significant impact on morbidity and mortality. Further, patients with liver cirrhosis, with or without AD, are at risk of progression to Acute-on-Chronic Liver Failure (ACLF) characterized by organ failures most often preceded or accompanied by an extreme inflammatory response and a poor survival [1,2]. 

A consensus definition of ACLF was introduced with the CANONIC-study conducted by the EASL Chronic Liver Failure Consortium including data from 1343 patients hospitalized in liver units throughout Europe. Main characteristics of ACLF comprise the following criteria [1]: (1) acute decompensation (rapid development of large-volume ascites, hepatic encephalopathy, variceal bleeding and/or bacterial infection); (2) organ failure (defined with the Chronic Liver Failure (CLIF)-SOFA-score; a modified version of the sequential organ failure assessment (SOFA) score including liver, kidney, cerebral, circulatory, respiratory and/or coagulation failure); (3) high 28-day mortality rate.

ACLF has an annual incidence among patients with compensated cirrhosis on 20.1 per 1000 persons [3]. The 28-day and 90-day mortality rates range between 22–77% and 41–79%, respectively, and dramatically increase with the number of organ failures [1]. A refined and simplified prognostic tool, the CLIF-C ACLF-score, including age and white blood cell count in addition to CLIF-organ-failure score, was later introduced and predict mortality with higher accuracy than CLIF-SOFA, Model for End-stage Liver Disease (MELD) and Child-Pugh-score [4].

Organ-specific and systemic inflammation are fundamental events in the development and course of chronic liver diseases [5,6] and macrophages play an important role for disease development and progression. Soluble CD163 and CD206 are promising biomarkers to reveal and quantify activation of the resident liver macrophages (Kupffer cells) [6], which have gained attention from a clinical perspective especially during the last decade.

New aspects of immunopathology in ACLF are highly relevant to support and extent current understandings. We intend to review the literature on macrophage activation as part of the immune response in liver diseases, especially in ACLF, with a focus on the soluble forms of CD163 and CD206 as biomarkers for ACLF disease severity and prognosis (Figure 1).

## 2. Macrophages

Macrophages were first recognized for their role in host immunity and phagocytosis, followed by evidence of their importance in development, tissue homeostasis, metabolism, and tissue regeneration [10,11]. Macrophages display a wide range of membrane receptors which recognize both host-derived and foreign ligands. Activation of these receptors can result in a pro-inflammatory response, leading to the recruitment and activation of other immune cells or an anti-inflammatory response in which the immune reaction is dampened and tissue regeneration and wound healing is promoted [10].

### Scavenger Receptors—Structure and Function

Scavenger receptors are highly expressed by macrophages [12]. These receptors typically bind a range of ligands and promote removal of non-self or altered-self targets. They often function through endocytosis, phagocytosis, adhesion, and signaling, resulting in elimination of degraded or harmful substances [12]. Two scavenger receptors highly expressed by macrophages are CD163 and CD206, also known as the hemoglobin-haptoglobin scavenger receptor and the mannose receptor, respectively [13,14].

Both CD163 and CD206 are expressed by macrophages but the expression of the receptors is differentially regulated. CD163 expression is increased in response to interleukin-10 (IL-10) stimulation, while CD206 expression is upregulated by IL-4 and IL-13 [11]. Macrophages are highly plastic cells, with the ability to switch their phenotype in response to external stimuli [15]. They have traditionally been classified as M1 ‘classical macrophages’ involved in bacterial and viral clearance and the release of pro-inflammatory cytokines (e.g., TNF, IL-1β, IL-12, and reactive oxygen species), or M2 ‘alternatively activated macrophages’ involved in defense against parasitic infections, tissue remodeling and secretion of immune-modulatory mediators (such as IL-10, TGF-β, IL-4, IL-13) [15]. However, recent advanced single-cell sequencing studies have demonstrated a much more differentiated activation of macrophages in inflammatory liver diseases as shown in non-alcoholic fatty liver disease (NAFLD) [16] and liver cirrhosis [17].

## 3. CD163

CD163 is a 130 kDa protein [18] exclusively expressed by monocytes and macrophages [18,19]. The main function of CD163 is to remove hemoglobin-haptoglobin complexes from the blood circulation during intravascular hemolysis [20]. CD163 is composed of a short intracellular domain, a single transmembrane segment, and an extracellular domain consisting of nine scavenger receptor cytokine-rich (SRCR) receptor class B domains [13] (Figure 2). Different isoforms of CD163 have been identified, including three splice variants in which the cytoplasmic tail differ in length [13]. While the CD163 short tail isoform is the most abundant, all variants display endocytic activity [21].

Hemoglobin-haptoglobin complexes bind extracellular domains in CD163, and the combined complex is subsequently endocytosed and degraded in the lysosomes [20,22]. However, CD163 is also involved in immunomodulatory functions such as immune-sensing of bacteria, binding of TNF-like weak inducer of apoptosis (TWEAK) and production of anti-inflammatory and anti-oxidative substances (IL-10, ferritin, bilirubin, CO) [23,24,25,26,27]. Furthermore, CD163 may be involved in erythroblast adhesion and promotion of erythroblast growth and survival [23,24,25,26,27].

### 3.1. CD163 Shedding

A soluble form of CD163, denoted sCD163, is present in plasma [28] and other bodily fluids [29] due to both constitutive and induced shedding. Soluble CD163 is therefore present in plasma from healthy individuals [28]. However, a range of stimuli, such as lipopolysaccharide (LPS), phorbol 12-myristate 13-acetate (PMA), and zymosan, can increase CD163 shedding both in vivo and in vitro [7,9,30,31].

Until recently sCD163 was believed solely to be released to the circulation by proteolytic cleavage of the membrane-bound form [7]. However, recent data showed that sCD163 exists in two forms, denoted Ecto-CD163 and EV-CD163 [8]. Ecto-CD163, is the dominant form and is released to the circulation after cleavage by tumor necrosis factor α (TNF-α) converting enzyme (TACE) [7]. The cleaved molecule comprises more than 94% of the CD163 ectodomain [32]. In contrast, EV-CD163 is a minor fraction of sCD163, at least in healthy individuals, released to the circulation as an extracellular vesicle-associated protein [8]. Although serum levels of total sCD163 has been extensively examined in a wide range of diseases [33,34,35,36,37,38,39,40,41], little is known about the distribution of the two sCD163 fractions in different clinical situations. The original study describing the existence of EV-CD163, found low levels, approximately 10%, of EV-CD163 in healthy individuals, while sepsis patients presented with EV-CD163 levels of up to 60% of total sCD163 [8]. In vivo, an LPS challenge in healthy individuals results in a fast increase in Ecto-CD163 within 1–2 h, thus it has been suggested that the two sCD163 fractions are released by different mechanisms and may constitute markers of different phases of the inflammatory response [8]. This is in agreement with unpublished data from our group in which we find an increased fraction of EV-CD163 in patients suffering from chronic alcoholic liver cirrhosis, compared to patients with alcoholic hepatitis. Lastly, we have evaluated the distribution of CD163 fractions in multiple myeloma in which we observed an increased fraction of EV-CD163 in newly diagnosed multiple myeloma patients compared to patients with relapse and in remission [42].

Interestingly, there appears to be differences in CD163 shedding between human and mice. The amino sequence recognized by TACE in humans is absent in murine CD163, thus CD163 shedding in mice is not mediated by TACE cleavage [43].

### 3.2. Soluble CD163 Function

To date, specific functions of sCD163 have not been established. Although the soluble protein retains its ability to bind hemoglobin-haptoglobin complexes, the affinity is much lower than membrane-bound CD163 [32]. The observed differences in affinity is likely because the hemoglobin-haptoglobin complex binding is increased with CD163 cross-linkage [32]. Further, data suggest a possible inhibitory effect on T-lymphocytes and binding of sCD163 to *Staphylococcus aureus* [44,45]. Extracellular vesicles (EVs) in general are thought to be important in intercellular communication, and macrophage derived EVs can alter the phenotype of endothelial cells, allowing an increased leucocyte recruitment and activation [46]. Along with this, these EVs have displayed the ability to activate specific T-cell populations [47]. Thus, although speculative, EV-CD163 may be involved in communication between macrophages and immune effector cells.

## 4. CD206

CD206 is a 175 kDa membrane-bound protein, primarily expressed by macrophages and dendritic cells, but also by lymphatic, hepatic, and splenic endothelium, kidney mesangial cells, tracheal smooth muscle cells, and retinal pigment epithelium [14,48,49,50]. CD206 is a complex molecule, comprising different extracellular domains, a transmembrane segment, and a cytoplasmic tail [51]. The extracellular part consists of an N-terminal cysteine rich (CR) domain, a fibronectin type II (FNII) domain, and eight C-type lectin domains (CTLDs) [51] (Figure 3). The receptor can undergo post-translational modifications, including glycosylation and conformational changes [52,53]. These modifications may affect ligand selectivity and binding affinity, as lack of terminal sialylation impairs binding and internalization of mannosylated carbohydrates through the CTLDs, while non-sialylation may increase CD206 aggregation, allowing an increased binding to sulfated ligands through the CR domain [53]. In addition, CD206 can adopt two different bend conformations, in which the CR and FNII domains are brought in proximity to CTLD3 and CTLD6, respectively [54]. These conformational changes appear to be pH-dependent, and may therefore play a role in ligand binding and release [54,55].

CD206 is involved in endogenous molecule clearance, antigen presentation, and modulation of cellular activity [56]. The extracellular part of CD206 allows for binding to sulfated carbohydrates through the CR domain, collagens through the FNII domain, and glycoconjugates terminated in mannose, fucose, or GlcNAc through the CTLDs [57,58,59,60,61]. Thus, CD206 recognizes and binds a wide range of ligands, including peptide hormones, lysosomal hydrolases, mannose, fucose, and collagen, along with allergens and microbial products including CpG DNA (a potent pathogen-associated immuno-modulatory component) [56,62,63,64,65,66,67,68]. In addition, CD206, like CD163, is an efficient endocytic receptor that continuously recycles between the cell surface and early endosomal compartments [69]. At steady state, as little as 10–30% of cellular CD206 is presented at the plasma membrane [69].

Despite the ability to recognize and bind pathogens, CD206’s contribution to host defense remains unclear. Although CD206 deficient mice display an impaired ability to remove collagen peptide hormones and lysosomal hydrolases [63], in vivo and in vitro studies indicated that CD206 alone is insufficient to induce phagocytosis, but may instead modulate signals induced by other receptors, such as Fc or Toll-like receptors [70].

### 4.1. CD206 Shedding

Like CD163, CD206 also exists in a soluble form, sCD206, but less is known about the mechanisms behind CD206 shedding [9,71,72]. Soluble CD206 is present in culture media from human dendritic cells, human macrophages, and murine macrophages, as well as in human and murine serum [9,72,73,74,75]. While sCD206 is present in plasma from healthy individuals, suggesting sCD206 production may be constitutive, the plasma concentration of sCD206 is increased in a wide range of diseases [74,75,76,77,78,79,80], along with in response to stimulation with fungi, LPS, and PMA in vivo and in vitro [9,71,74].

Studies have found that the soluble form of CD206 is smaller than the membrane-bound version and comprises the extracellular domain including the CR domain, the FNII domain, as well as all of the CTLDs [75]. In mice, CD206 shedding is increased in response to fungal stimuli through Dectin-1 engagement and has been suggested to be mediated by matrix metalloproteases [71,81]. To date, only one study has evaluated the shedding mechanism behind CD206 release in humans [9]. Here we showed that although CD206 shedding in humans appear to be protease mediated, the release in humans is not mediated through the engagement of matrix metalloproteases unlike the shedding in mice [9].

Additionally, we have presented data suggesting that CD206, like CD163, also exists in an extracellular vesicle-associated form (Unpublished data and [42]). Using ExoQuick™ as the means of EV-isolation, we were able to measure sCD206 in both the free protein and the EV-associated protein fraction. While further studies are required to confirm the extracellular vesicle association of this soluble protein, the data strongly suggests the existence of EV-associated CD206.

### 4.2. sCD206 Function

The specific function of sCD206 is unknown, but similar to sCD163, the soluble receptor retains its ability to bind ligands. Soluble CD206 may therefore play a role in antigen capture and antigen transport to cells involved in the humoral immune response [82]. Specifically, sCD206 released from macrophages has been hypothesized to bind and transport CTLD ligands to CR ligand expressing macrophages in the marginal zone of the spleen and subcapsular sinus of the lymph nodes, thereby possibly generating an immune response [82].

## 5. sCD163 and sCD206 in Liver Disease

Liver cirrhosis is the end-stage of ongoing inflammation and fibrosis in chronic inflammatory liver diseases. Liver inflammation and fibrosis may progress at very different time frames with slow progression in NAFLD, primary biliary cholangitis, and chronic viral hepatitis B (HBV) and C (HBC), however, rapid progression may occur in some patients. More aggressive inflammation and fibrosis are seen in alcoholic liver disease especially alcoholic hepatitis and untreated autoimmune hepatitis. In all chronic liver diseases Kupffer cells and recruited macrophages are involved in the inflammatory process. Further, the macrophages play a key role in the acute deterioration in patients with ACLF. Since sCD163 and sCD206 may be used as specific markers of macrophage activation, we and others have investigated sCD163 and sCD206 in acute and chronic liver diseases including ACLF. 

### 5.1. sCD163 in Liver Diseases

Studies on sCD163 and sCD206 showed increased levels in relation to disease severity and prognosis in as well acute liver failure as chronic inflammatory liver diseases as shown in Figure 4 and Figure 5. Slight to moderate elevations of plasma sCD163 are observed in obese adults with biopsy verified NAFLD and non-alcoholic steatohepatitis (NASH) and the plasma concentration reflects the severity of NAFLD [83,84,85] and levels are reduced after interventions [86,87]. In chronic HBV and HCV infection sCD163 levels increase with incrementing stages of liver inflammation and fibrosis and significant reductions are observed after antiviral therapy. [35,88,89,90,91]. Further, a sCD163 based HCV fibrosis score is a better predictor for liver fibrosis than the traditional FIB4 and APRI scores [35]. The sCD163 concentration is significantly elevated in patients with liver cirrhosis and associated with liver disease severity as determined by the Child-Pugh-score and MELD [92,93,94]. A gradient across the liver has been demonstrated suggesting intrahepatic secretion [95]; and the sCD163 concentration correlates with the degree of portal hypertension, and represents a clinically relevant non-invasive and cost-effective tool to detect clinically significant portal hypertension in cirrhosis patients [92,95]. Furthermore, the sCD163 level is independently associated with variceal bleeding in cirrhosis patients [94]. Marked elevation of sCD163 has been observed in patients with alcoholic hepatitis and a high sCD163 concentration is a predictor of mortality [33,96]. Highest levels of sCD163 are described in patients with acute liver failure especially among patients with fatal outcome [34]. Summarized, the sCD163 concentration is a useful marker of the degree of inflammation especially in liver diseases, due to shedding of sCD163 from the cell surface of activated macrophages [29,97]. Thus, a correlation between sCD163 level and liver disease severity is evident [97,98] and the Kupffer cell activation marker has, in multiple studies, showed to be a potential independent predictor of mortality [1,33,79,94,96,97].

### 5.2. sCD206 in Liver Diseases

Similar to sCD163, sCD206 has gained increasing interest due to its potential as a reliable biomarker of ongoing inflammation. Concentrations of sCD206 correlate with concentrations of sCD163 reflecting that both biomarkers are shed from activated macrophages, though signals for shedding of sCD163 and sCD206 differ, as discussed above [75]. Clinical studies of sCD206 in patients with chronic HCV and HBV showed significant associations with inflammation and fibrosis severity determined by Ishak fibrosis score or transient elastography [80,100]. Further, significant persistent reductions in sCD206 concentrations are observed after antiviral treatment [80,91]. Significantly elevated sCD206 are present in patients with cirrhosis compared to healthy individuals, and increase with increased liver disease severity including decompensation with ascites [78,79]. Moreover, sCD206 concentrations predict survival, and plasma levels are significantly higher in patients experiencing cirrhosis complications [79]. Another study in cirrhosis patients found a correlation between portal and hepatic vein sCD206 concentrations and the portal pressure prior to insertion of a transjugular intrahepatic portosystemic shunt (TIPS), and with a gradient across the liver [101]. High concentrations are observed in patients with acute liver injury due to an acetaminophen overdose [99] and highest concentrations are found in patients with alcoholic hepatitis [75,78,96]. The alcoholic hepatitis patients had higher numbers of peripheral monocytes compared to healthy controls and alcohol cirrhosis patients. However, there was no expression of membrane bound CD206 on CD14^+^ monocytes in neither patients nor controls, which suggest sCD206 to derive from the liver and not peripheral blood monocytes [78]. Among patients with alcoholic cirrhosis, the sCD206 level predicts portal hypertension and 4-year mortality [78]. As illustrated in Figure 5, the sCD206 concentration increases in relation to the liver diseases severity and mimics the sCD163 results displayed in Figure 4. Recently, sCD206 has been quantified in ascites fluid from cirrhosis patients with spontaneous bacterial peritonitis. The ascites fluid sCD206 concentration is a marker of peritoneal macrophage activation and peritoneal inflammation, and may be a predictor of reduced 90-day survival [102]. 

## 6. Acute-on-Chronic Liver Failure

Inflammation and toll-like receptor activation either from infections and pathogen-associated molecular patterns (PAMPs) or by cellular decomposition and production of damage-associated molecular patterns (DAMPs) seems to play a key role in AD and ACLF development and progression; and may result in significant macrophage activation and an exaggerated immune response [5,103].

Precipitating events of ACLF, either hepatic or extra-hepatic, are identified in approximately 60% of patients but with large variations in cause and incidence between continents. Events of hepatic origin includes high alcohol intake, viral hepatitis, ischemic hepatitis, drug-induced liver injury, liver surgery, and insertion of a TIPS, while extrahepatic causes include bacterial infections, surgery, and paracentesis without volume expansion [1,5]. Though the type of precipitating event has no relation to mortality [1] the characterization of such events may be crucial to understand the pathophysiology as well as histo- and immuno-pathogenesis of ACLF, additionally explaining the diversity of former and present understandings and definitions of the condition [104].

In ACLF patients, blood biochemistry reveals significant elevations of C-reactive protein, white blood cell count and pro-inflammatory cytokines and chemokines [1,105,106]. These findings parallel the well-established understanding regarding induction of an excessive systemic inflammatory response in ACLF [5,103]. Note, as a marker of ongoing inflammation, white blood cell count is part of the CLIF-C ACLF-score [4]. Furthermore, plasma concentrations of sCD163 and sCD206, as described above, are useful indicators of ongoing inflammation and fibrosis in chronic liver diseases. Thus, elevation of sCD163 and sCD206 may as well be clinically relevant for evaluation of inflammation in ACLF patients and potentially used to predict mortality [6]. 

### 6.1. sCD163 in ACLF

Only a few studies have focused on sCD163 in AD and ACLF. In a retrospective study restricted to HBV infected patients who progressed to acute liver failure, significant elevations of sCD163 were observed compared with both chronic HBV patients and healthy controls [107]. Grønbæk et al. [6] investigated sCD163 levels from 851 patients included in the CANONIC-study. A stepwise dramatic increase in sCD163 concentration was observed with increasing ACLF grades (Figure 4). Furthermore, sCD163 was independently associated with 28-day and 90-day mortality and significantly improved the prediction of 90-day mortality if added to the original CLIF-C ACLF-score. In non-surviving ACLF patients a significant increase in sCD163 from baseline to day 3–7 was observed compared to survivors. Another interesting finding was higher sCD163 levels in ACLF patients with liver, coagulation, or cardiovascular failure. Since PAMPs are considered crucial for the development of excessive systemic and local inflammation with macrophage activation in ACLF and other liver diseases, high levels of sCD163 were, not surprisingly, documented in patients with bacterial infection and sepsis as precipitating events [6]. Just recently, a significant association was observed between sCD163 (and sCD206) and a 38-metabolite blood fingerprint associated with an intense metabolic derangement characterized by enhanced proteolysis, lipolysis, and aminoacid catabolism and impaired mitochondrial function in peripheral organs, which may contribute to organ failure development in ACLF [108]. Similar findings were demonstrated in a HBV ACLF subpopulation of Chinese patients where only approximately 50% had established cirrhosis [109]. They reported significantly higher sCD163 levels in advanced ACLF compared with early (no organ failure) ACLF and in ACLF patients with failure of the coagulation system. Further, they observed an independent association between sCD163 and 28-day mortality, and improvement of traditional prognostic scores with the incorporation of sCD163 [109]. However, sCD163 concentrations overall were much lower, without a stepwise increase with increasing disease severity and without significant difference in sCD163 concentration in patients with bacterial infections compared with uninfected patients [109]. This incongruence may be explained by fewer and less systemically inflamed patients, only HBV-associated ACLF, exclusion of severe extrahepatic disease, and lack of a non-ACLF control group [109]. In a previous study sCD163 levels in pneumonia patients were elevated compared to controls, however, the presence of underlying alcoholic liver disease resulted in significantly higher sCD163 levels [110]. Thus, extrahepatic infections e.g., pneumonia may trigger the constitutive liver macrophage activation in liver cirrhosis and result in amplified liver macrophage activation accompanied by an exaggerated immune response and potential cytokine storm leading to organ dysfunction and ACLF during e.g., pneumonia. 

### 6.2. sCD206 in ACLF

sCD206 was in addition to sCD163 quantified in serum from 851 patients included in the CANONIC-study by Grønbæk et al [6]. With increasing disease severity, a clear stepwise increase in sCD206 concentration, similar to sCD163, was documented (Figure 5). Additionally, sCD206 was associated with 28-day and 90-day mortality in ACLF patients but with a prognosis prediction accuracy slightly inferior to MELD and CLIF-C ACLF-score. However, combining sCD206 with the CLIF-C AD-score, a specific score for hospitalized AD patients without ACLF [111], significantly improved the prediction of 90- and 180-days mortality in non-ACLF patients [6]. In relation to organ failure, the sCD206 level was significantly higher in ACLF patients experiencing liver, coagulation, cardiovascular or cerebral failure. The latter is beyond the findings with sCD163. Like sCD163, an increase in sCD206 from baseline to day 3–7 was significant in non-surviving ACLF patients compared with survivors, and high levels of sCD206 were present in patients with bacterial infection and sepsis as precipitating events [6]. Conversely, in sepsis patients the highest sCD206 levels were observed in those with underlying liver disease [75]. Recently, a study on 43 patients with HBV associated ACLF was published, which classified disease stages according to the grade of coagulation system dysfunction [112]. The sCD206 concentration gradually increased with increasing coagulopathy [112]. Consistent with findings in the CANONIC cohort, sCD206 was significantly increased among non-survivors compared with survivors, and sCD206 showed potential as a predictor of mortality since addition of the biomarker to the MELD score significantly improved the prognostic accuracy [112]. Similar to sCD163, sCD206 was associated with an intense metabolic derangement in these patients [108].

### 6.3. Similarities and Differences in sCD163 and sCD206 in AD and ACLF

Certainly, macrophage activation is essential for development and maintenance of an inflammatory response in liver diseases and particularly in AD and ACLF patients. Results on sCD163 and sCD206 in AD and ACLF obtained from the CANONIC cohort ascertain prognostic values of both biomarkers [6] as confirmed by others [109,112].

Recently, a battery of inflammatory mediators, including factors of macrophage stimulation as well as monocyte migration and macrophage activation, was quantified in different degrees of acute decompensated cirrhosis, and the intensity of baseline inflammation may predict risk of disease progression to ACLF and mortality [113]. Moreover, all grades of ACLF display a clearly different blood metabolite fingerprint compared with AD patients [108]. Thus, although characteristics of the inflammatory drivers activated in AD and ACLF overlap, the sepsis-like inflammation in ACLF, triggered by precipitating events, may be syndrome-specific [5,114]. Different activators of CD163 and CD206 expression on the macrophage surface as well as different shedding mechanisms of the soluble forms may explain dissimilarities in prognostic potentials of sCD163 and sCD206 in AD and ACLF patients.

Paralysis of monocytes in response to PAMPs in ACLF has been conceptualized, and data suggest a change in monocyte function [114] including elevated frequencies of IL-10-producing monocytes [115]. IL-10 upregulates CD163 on macrophages [11] and significantly elevated plasma concentrations of IL-10 are present in ACLF [106]. Further, a significant expansion of macrophage CD163 expression is evident in liver tissue from ACLF patients [105] and CD163 shedding from Kupffer cells is stimulated by well-characterized shedding-factors including both DAMPs and PAMPs through Toll-like receptors [9]; both associated with induction of ACLF and disease severity [5,116]. These findings are in congruence with the fact that sCD163 reaches extremely high plasma levels in ACLF, strengthening the theoretical fundament of sCD163 as a key prognostic biomarker in ACLF, and as evident by improvement of the 90-day mortality prediction when added to the CLIF-C ACLF-score [6].

Similar to sCD163, shedding of sCD206 is promoted by PAMPs including LPS. However, sCD206 release occurs significantly slower than sCD163 in response to LPS [9]. Further, sCD206 shedding is induced by other stimulants and parallels with a less pronounced and slower release than sCD163 [9]. These shedding-characteristics indicate a different immunogenic role of sCD206, which may explain why the supplementation of sCD206 to the CLIF-C AD-score significantly improves 90-day and 180-day mortality prediction among AD (non-ACLF) individuals [6,111]. Finally, it is tempting to hypothesis that distribution patterns of the sCD163 and sCD206 forms reflect responses of different immunological conditions. However, future clinical studies are required to test the hypothesis.

## 7. Conclusions

Here we reviewed the literature on macrophage scavenger receptor functions in inflammatory liver diseases with focus on sCD163 and sCD206 and patients with ACLF. These patients are facing a poor prognosis as treatment options are limited to treatment of identified precipitating events and supportive care. Ultimately, liver transplantation must be considered [5]. Thus, new perspectives of the syndrome and pathogenesis including inflammation markers are crucial to advance medico-pharmaceutical measures. Clinical evidence demonstrates a robust association of the novel biomarkers sCD163 and sCD206 with inflammation in liver diseases, and especially ACLF patients produce exceedingly high plasma sCD163 and sCD206 levels, and both definitely associated with mortality. Further, the markers exhibit a potential to separate survivors and non-survivors within the first week of diagnosis, which may help decision making for more aggressive treatments and follow-up and may even be target for intervention. Future studies will determine where and when the biomarkers may be incorporated in the clinical work, though it is already evident that well-established scoring tools may be strengthened by addition of sCD163 and/or sCD206.

## Figures and Tables

**Figure 1 cells-09-01175-f001:**
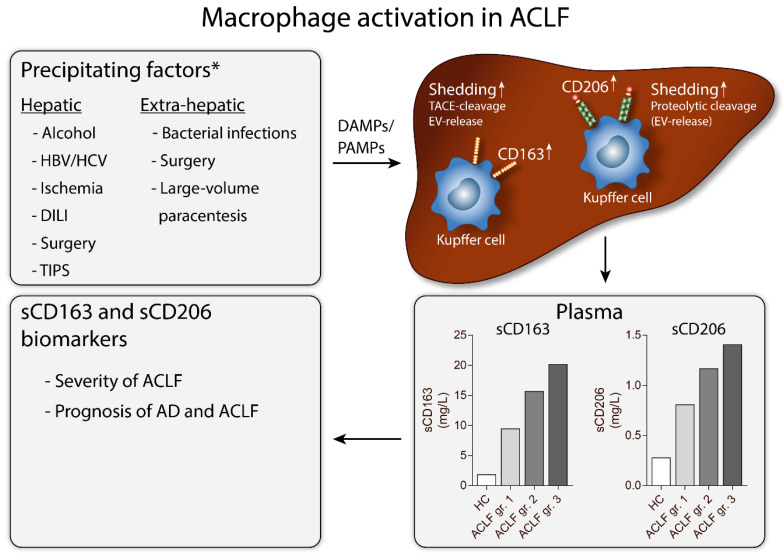
Macrophage activation in patients with acute-on-chronic liver failure (ACLF). Hepatic or extra-hepatic precipitating factors result in the release and production of PAMPs and DAMPs. This leads to increased CD163 and CD206 expression in liver resident macrophages (Kupffer cells), along with increased shedding of the receptors from the Kupffer cells. CD163 shedding is mediated through TACE-cleavage [7] and EV-associated release [8], whereas CD206 shedding is mediated through proteolytic cleavage [9] and possibly EV-associated release (unpublished data). The relationship between CD163 and CD206 proteolytic cleavage and EV-associated release is not fully elucidated. Plasma concentrations of soluble CD163 and CD206 correlates with the severity of ACLF and are promising prognostic biomarkers in AD and ACLF [6]. Abbreviations: HBV, hepatitis B virus; HCV, hepatitis C virus; DILI, drug-induced liver injury; TIPS, transjugular intrahepatic portosystemic shunt; DAMPs, damage-associated molecular patterns, PAMPs, pathogen-associated molecular patterns; TACE, tumor necrosis factor α converting enzyme; EV, extracellular vesicle; HC, healthy controls; ACLF, Acute-on-Chronic Liver Failure; gr., grade; AD, acute decompensation. * Precipitating factors as presented in [1,5].

**Figure 2 cells-09-01175-f002:**
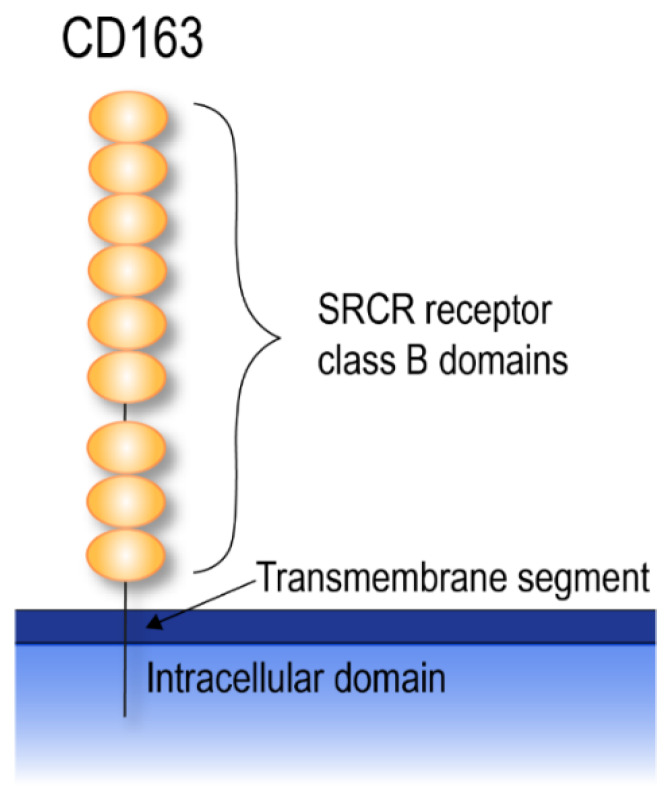
CD163 structure. CD163 is composed of a short intracellular domain, a single transmembrane segment, and an extracellular domain consisting of nine SRCR receptor class B domains. SRCR, scavenger receptor cytokine-rich.

**Figure 3 cells-09-01175-f003:**
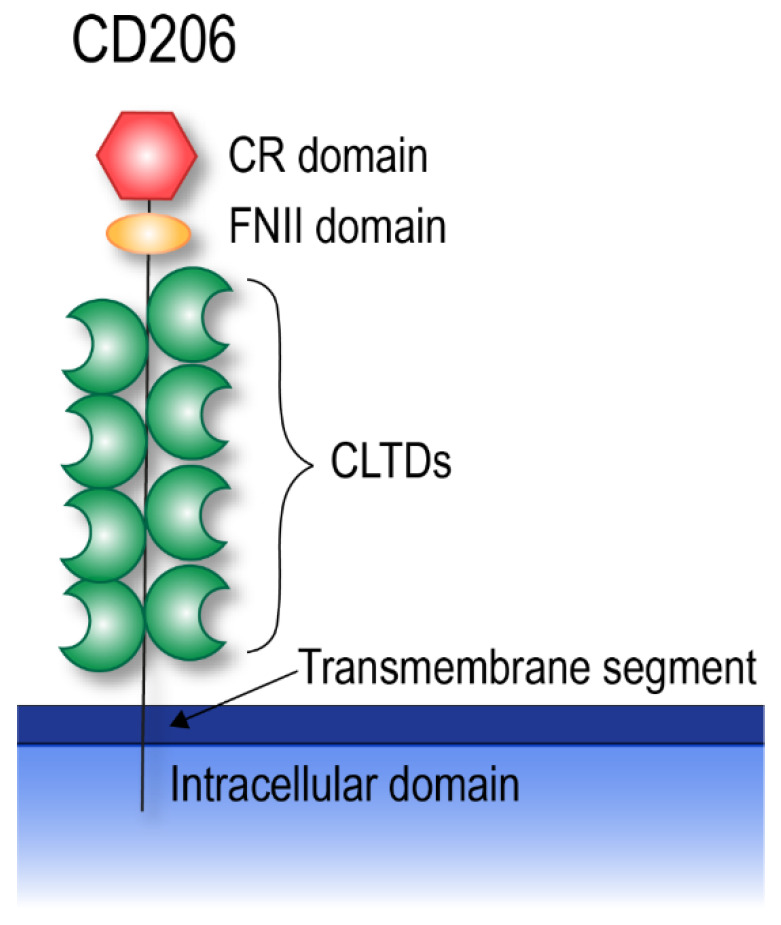
CD206 structure. CD206 is composed of an intracellular domain, a single transmembrane segment, and an extracellular domain. The extracellular domain is comprised of an N-terminal CR domain, a FNIII domain, and eight CTLDs. CR, cysteine rich; FNII, fibronectin type II; CTLD, C-type lectin domain.

**Figure 4 cells-09-01175-f004:**
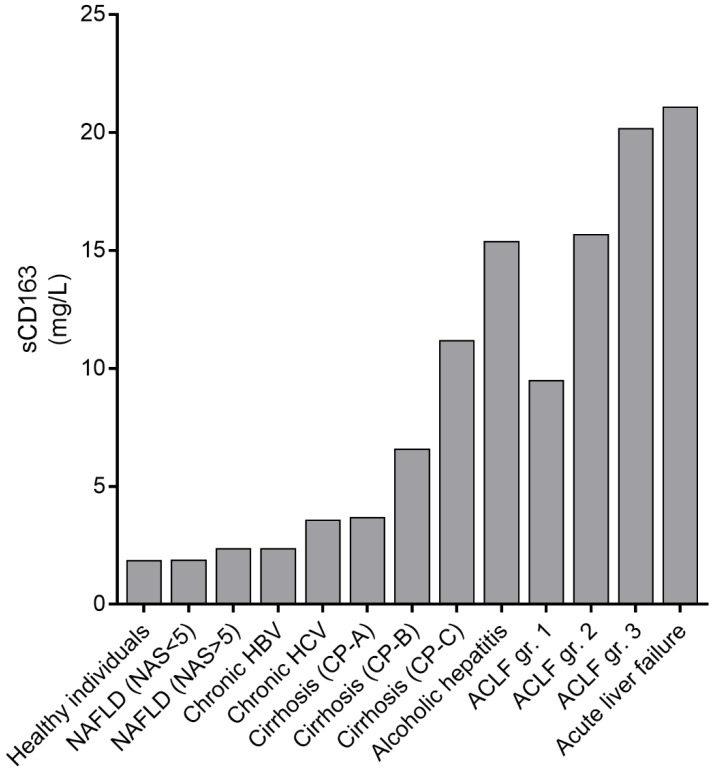
Median concentrations of sCD163 from selected studies covering healthy individuals and patients with liver diseases of different etiology and increasing liver disease severity [6,28,33,34,35,83,93]. NAFLD, non-alcoholic fatty liver disease; NAS, NAFLD activity score; HBV, hepatitis B virus; HCV, hepatitis C virus; CPA, Child-Pugh-score A; CPB, Child-Pugh-score B; CPC, Child-Pugh-score C; ACLF, Acute-on-Chronic Liver Failure; gr., grade.

**Figure 5 cells-09-01175-f005:**
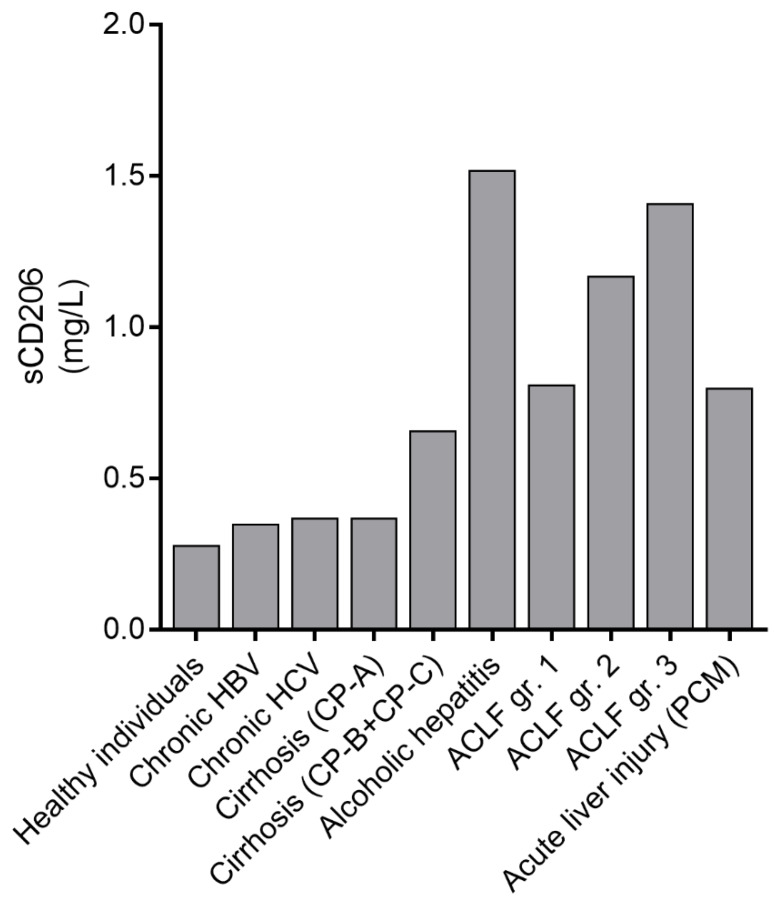
Median concentrations of sCD206 from selected studies covering healthy individuals and patients with liver diseases of different etiology and increasing disease severity [6,75,78,79,80,91,99]. HBV, hepatitis B virus; HCV, hepatitis C virus; CPA, Child-Pugh-score A; CPB, Child-Pugh-score B; CPC, Child-Pugh-score C; PCM, paracetamol (acetaminophen); ACLF, Acute-on-Chronic Liver Failure; gr., grade.

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
