# Peer review of "Macrophage Activation Markers, CD163 and CD206, in Acute-on-Chronic Liver Failure"

_cells, 2020, doi:10.3390/cells9051175_

Round 1

Reviewer 1 Report

Review Article: Macrophage activation markers, sCD163 and sCD206, in Acute-on-Chronic Liver Failure

Text - comments:

Increased surface CD163 expression on peripheral monocytes from patients with ACLF and ALF has been reported (Bernsmeier et al. 2015; Triantafyllou et al. 2018). These studies have also shown, by tissue immunohistochemistry, an expansion of CD163+ macrophages in liver tissue of patients with ACLF and ALF. Maybe worthwhile to mention in text as a potential source with regards to increased soluble CD163 plasma levels? Similarly, is there any evidence for increased surface CD206 expression of monocytes in ACLF? A recent study Stengel et al. showed that increased peritoneal levels (ascitic fluid) of soluble CD206, released by activated large peritoneal macrophages, associate with patient mortality in decompensated cirrhosis and spontaneous bacterial peritonitis (Stengel et al. 2020); maybe relevant to ACLF?

Figures - comments:

Figure 1: The data graphs (sCD163 and sCD206) included in bottom right square of the figure need to be bigger in size, and increase their resolution, as can’t be easily read/followed (it’s HC vs different grade ACLF).

References:

Bernsmeier, C., et al. 2015. “Patients with Acute-on-Chronic Liver Failure Have Increased Numbers of Regulatory Immune Cells Expressing the Receptor Tyrosine Kinase MERTK.” Gastroenterology 148 (3). https://doi.org/10.1053/j.gastro.2014.11.045.

Stengel, Sven, Stefanie Quickert, et al. 2020. “Peritoneal Level of CD206 Associates With Mortality and an Inflammatory Macrophage Phenotype in Patients With Decompensated Cirrhosis and Spontaneous Bacterial Peritonitis.” Gastroenterology. https://doi.org/10.1053/j.gastro.2020.01.029.

Triantafyllou, Evangelos, et al. 2018. “MerTK Expressing Hepatic Macrophages Promote the Resolution of Inflammation in Acute Liver Failure.” Gut 67 (2): 333–47. https://doi.org/10.1136/gutjnl-2016-313615.

Author Response

Reviewer 1 

Text - comments:

Increased surface CD163 expression on peripheral monocytes from patients with ACLF and ALF has been reported (Bernsmeier et al. 2015; Triantafyllou et al. 2018). These studies have also shown, by tissue immunohistochemistry, an expansion of CD163+ macrophages in liver tissue of patients with ACLF and ALF. Maybe worthwhile to mention in text as a potential source with regards to increased soluble CD163 plasma levels?

Similarly, is there any evidence for increased surface CD206 expression of monocytes in ACLF? A recent study Stengel et al. showed that increased peritoneal levels (ascitic fluid) of soluble CD206, released by activated large peritoneal macrophages, associate with patient mortality in decompensated cirrhosis and spontaneous bacterial peritonitis (Stengel et al. 2020); maybe relevant to ACLF?

Answer: Thanks for pointing out these important references. We have included them in the revised manuscript in relevant sections.

In a recent study we investigated CD206 expression on blood monocytes in patients with alcoholic hepatitis. There was an increase in the numbers of peripheral monocytes in these patients compared to both the healthy controls and alcohol cirrhosis patients (p = 0.006); However, there was no expression of membrane bound CD206 on CD14+ monocytes in neither patients nor controls. This has been included in the revised manuscript.

Figures - comments:

Figure 1: The data graphs (sCD163 and sCD206) included in bottom right square of the figure need to be bigger in size, and increase their resolution, as can’t be easily read/followed (it’s HC vs different grade ACLF).

Answer: Thanks – we have improved the figures.

References:

Bernsmeier, C., et al. 2015. “Patients with Acute-on-Chronic Liver Failure Have Increased Numbers of Regulatory Immune Cells Expressing the Receptor Tyrosine Kinase MERTK.” Gastroenterology 148 (3). https://doi.org/10.1053/j.gastro.2014.11.045.

Stengel, Sven, Stefanie Quickert, et al. 2020. “Peritoneal Level of CD206 Associates With Mortality and an Inflammatory Macrophage Phenotype in Patients With Decompensated Cirrhosis and Spontaneous Bacterial Peritonitis.” Gastroenterology. https://doi.org/10.1053/j.gastro.2020.01.029.

Triantafyllou, Evangelos, et al. 2018. “MerTK Expressing Hepatic Macrophages Promote the Resolution of Inflammation in Acute Liver Failure.” Gut 67 (2): 333–47. https://doi.org/10.1136/gutjnl-2016-313615.

Reviewer 2 Report

The manuscript is a review on the role of sCD163 and sCD206, compounds secreted by macrophages, as predictors of morbidity and mortality of acute and chronic liver diseases. The study collects the literature on the subject in an updated and easily understandable way for the reader. The update helps to know which markers of liver inflammation could be taken into account in the future for the prognosis of liver inflammation.

Author Response

Reviewer 2 

Comments and Suggestions for Authors

The manuscript is a review on the role of sCD163 and sCD206, compounds secreted by macrophages, as predictors of morbidity and mortality of acute and chronic liver diseases. The study collects the literature on the subject in an updated and easily understandable way for the reader. The update helps to know which markers of liver inflammation could be taken into account in the future for the prognosis of liver inflammation.

Answer: Thanks for the positive comments

Reviewer 3 Report

In this manuscript, Marlene Christina Nielsen et al reviewed the structure, function, and shedding mechanism of macrophage activation markers CD163 and CD206. Furthermore, the authors critically reviewed the potential clinical utility of these proteins in the context of acute to chronic liver diseases with a particular focus on ACLF.

Overall, the review manuscript is properly designed and well written considering both the clinical and basic aspects of the macrophages markers. The review is short yet comprehensive and easy to follow.

Some minor comments are as below-

  • Schematic representation of structural motifs for both CD163 and CD206 would make the appearance of the manuscript better.
  • In Figure 1: which particular hepatic cell show increased shedding? What is the relationship of increase shedding to the EV release, and proteolytic cleavage for both CD163 and CD206? The figure doesn't convey these critical conceptual messages.
  • Figures 3 and 4: The tabular design should be made uniform with a visible x- and y-axis line.

Author Response

Reviewer 3 

In this manuscript, Marlene Christina Nielsen et al reviewed the structure, function, and shedding mechanism of macrophage activation markers CD163 and CD206. Furthermore, the authors critically reviewed the potential clinical utility of these proteins in the context of acute to chronic liver diseases with a particular focus on ACLF.

Overall, the review manuscript is properly designed and well written considering both the clinical and basic aspects of the macrophages markers. The review is short yet comprehensive and easy to follow.

Some minor comments are as below-

  • Schematic representation of structural motifs for both CD163 and CD206 would make the appearance of the manuscript better.

Answer: We aggree and have have included these figures in the revised version of the MS.

  • In Figure 1: which particular hepatic cell show increased shedding? What is the relationship of increase shedding to the EV release, and proteolytic cleavage for both CD163 and CD206? The figure doesn't convey these critical conceptual messages.

Answer: Thanks for the comment. We have tried to improve the figure as suggested and added further information in the legend.

  • Figures 3 and 4: The tabular design should be made uniform with a visible x- and y-axis line.

Answer: We agree and have revised the design